# PROBABILISTIC NEURAL-SYMBOLIC MODELS FOR INTERPRETABLE VISUAL QUESTION ANSWERING

## ABSTRACT

We propose a new class of probabilistic neural-symbolic models for visual question answering (VQA) that provide interpretable explanations of their decision making in the form of programs, given a small annotated set of human programs. The key idea of our approach is to learn a rich latent space which effectively propagates program annotations from known questions to novel questions. We do this by formalizing prior work on VQA, called module networks (Andreas, 2016) as discrete, structured, latent variable models on the joint distribution over questions and answers given images, and devise a procedure to train the model effectively. Our results on a dataset of compositional questions about SHAPES (Andreas, 2016) show that our model generates more interpretable programs and obtains better accuracy on VQA in the low-data regime than prior work.

## 1 INTRODUCTION

Exciting progress has been made in recent years in deep representation learning (Hochreiter & Schmidhuber, 1997; LeCun et al., 1998), which has led to advances in artificial intelligence tasks such as image recognition (Krizhevsky et al., 2012; He et al., 2015), machine translation (Sutskever et al., 2014), visual question answering (Agrawal et al., 2015), visual dialog (Das et al., 2016), and reinforcement learning (Mnih et al., 2015), *etc.* While deep neural networks achieve such impressive performances, many aspects of human cognition such as compositional generalization and reasoning are harder to model with state-of-the art deep learning approaches (Battaglia et al., 2018; Barrett et al., 2018).

Symbolic approaches (Newell & Simon, 1976) on the other hand provide strong compositional and reasoning capabilities which are challenging to model with neural networks (Lake et al., 2017). Consequently, a rich line of work in neuro-symbolic processing sought to build AI systems with both strong *learning and reasoning* capabilities (Valiant, 2003; Evans et al., 2018b; Bader & Hitzler, 2005).

As we tackle higher-level tasks which involve reasoning, such as visual question answering (Agrawal et al., 2015), planning (Andreas et al., 2016b; Gregor et al., 2016) *etc.*, it is natural to desire the ability to provide instructions to models to guide them. Symbolic instructions, by their very nature are easier to specify and more interpretable than specifying the parameters of a neural network. For such high-level tasks, a sensible approach is to specify "what" to do using symbols and learn how to do the task using modern representation learning techniques.

For the example shown in Figure 1, given a question "*Is a square to the left of a green shape?*", one can ask the model to reason about the answer by first applying a `find[green]` operator, then `find[left]`, then `And` the result together with a `find[square]` operator in order to predict the answer, in other words specifying "what" are the computations we desire to be executed, in the form of a "program". (Johnson et al., 2017; Hu et al., 2017; Andreas et al., 2016a). The network can then learn "how" to execute such a program from data using deep representation learning (Johnson et al., 2017; Hu et al., 2017; Andreas et al., 2016a).

This paper addresses a very natural desiderata for such neuro-symbolic models, in the context of visual question answering – can we retain the intepretability of "what the network did" expressed in terms of the syntax and lexicon that we are interested in, while specifying minimal teaching examples of "what to do" given an input question?

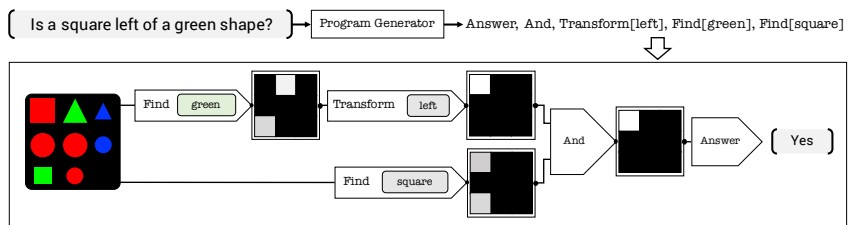

Figure 1: An illustration of neural module networks for visual question answering. Given a question, the program generator produces a program in prefix notation which is used to construct a neural network from the specified module networks. This network operates on the image and intermediate attention maps to answer the question.

We propose to approach this problem with a treatment of programs $\mathbf{z}$ as a latent variable, embedded into a generative model of questions $\mathbf{x}$ and answers $\mathbf{a}$ given images $\mathbf{i}$ as visualized in Figure 2. This model class has certain desirable properties: first, a proper treatment of $\mathbf{z}$ as a stochastic latent variable is helpful especially in the presence of a limited number of expert human instructions, as we model the uncertainty associated with $\mathbf{z}$ explicitly, leading to more interepretable latent spaces (Section 5); second, the model makes intuitive independence assumptions with regards to the question answering task – conditioned on the program $\mathbf{z}$, questions $\mathbf{x}$ are independent of the answers $\mathbf{a}$ for an image, meaning that the program must be a sufficient statistic for predicting the answer given a question[1]. Third, we parameterize the $p(\mathbf{a}|\mathbf{i}, \mathbf{z})$ term, which predicts an answer given an image and a program using a popular, related class of question answering models called neural module networks (NMNs) (Hu et al., 2017; Johnson et al., 2017; Andreas et al., 2016a) which previous work has shown leads to better fit between the program (or "what the model is doing") and the execution (or "how it is doing what it is doing"). Given the close ties of our model to NMNs, we will refer to our model as variational neural module networks (V-NMN) for the rest of the paper.

Apart from our modeling contribution of a probabilistic latent variable, neuro-symbolic model, our key technical contribution is to show how to train this model in the context of visual question answering. Optimizing such generative models with structured discrete latent variables is an intractable problem to solve in genereal – for this particular case, we formulate training into stages of grounding questions into programs (question coding), learning how to execute them and then with those initializations train the full variational objective. We demonstrate that this stage-wise optimization allows us to successfully learn a probabilistic neural-symbolic model that can answer compositional questions about shapes in an image (Andreas et al., 2016a).

While the V-NMN model is instantiated for the specific problem of visual question answering, we believe the tools and techniques we develop are more generally applicable to probabilistic, structured, discrete, interpretable latent spaces for interpretable neuro-symbolic models for planning, instruction following, grounding referring expressions etc.

We benchmark our model on the SHAPES (Andreas et al., 2016a) dataset with compositionally novel questions and show that our model is able to provide interpretable explanations of "what it is doing" (measured by program correctness) for unseen questions even in the setting where it is given as few as $\approx 20$ paired examples of programs and corresponding questions from the dataset, while also getting to high question answering accuracy. We find that the proposed approach outperforms a state of the art neural-symbolic model designed for VQA from Johnson et al. (2017), as well as deterministic ablations of our model.

## 2    A PROBABILISTIC NEURAL-SYMBOLIC MODEL FOR VQA

For a standard neural-symbolic NMN approach to question answering  (Johnson et al., 2017), we are given an input image $\mathbf{i}$ and a question $\mathbf{x} = (x_1, \cdots, x_N)$ as a sequence of words $x_i$ each encoded using a vocabulary $\mathcal{V}_{\mathbf{x}}$, and must produce answers $\mathbf{a}$ by planning and executing a series of modular

---

[1] Note that this model assumes independence of programs from images, which corresponds to the weak sampling assumptions in concept learning (Tenenbaum, 1999), one can handle question premise, ie. that people might ask a specific set of questions for an image in such a model by reparameterizing the answer variable to include a relevance label.

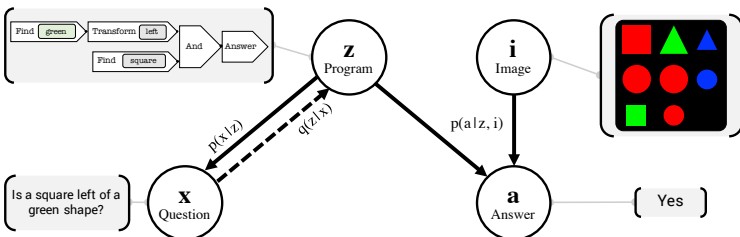

Figure 2: An illustration of the graphical model used for neural module networks (NMNs).

computations ($\mathbf{z} = (z_1, \cdots, z_N)$) – *i.e.* a program of operations ($z_i \in \mathcal{V}_\mathbf{z}$) to be executed in order to answer the question. Note that $\mathbf{z}$ here is the prefix serialization of the program and programs themselves often have tree-structured topologies. We denote the dataset of ($\mathbf{i}, \mathbf{x}, \mathbf{a}$) triplets as $\mathcal{D}$. Additionally, we are provided programs $\mathbf{z}$ for some input questions ($\mathbf{x} \in \mathcal{S}$), which specify the notion of grounding desired by an end user. Our goal is to provide faithful explanations on unseen questions (along with correct answers) consistent with the specified grounding.

We accomplish this by modeling $\mathbf{z}$ as a latent variable that underlies both question and answer generation given an input image. Intuitively, we expect this will enable us to learn a more meaningful latent representation for questions, as modeling uncertainty should help better handle the potentially underspecified mapping between questions and programs provided by the user. As such, our model factorizes $p(\mathbf{x}, \mathbf{z}, \mathbf{a}|\mathbf{i}) = p(\mathbf{x}|\mathbf{z})p(\mathbf{a}|\mathbf{i}, \mathbf{z})p(\mathbf{z})$ as in the graphical model shown in Figure 2, where the prior $p(\mathbf{z})$ is learnt by training a neural sequence model on valid strings simulated from known syntax. Conceptually, this factorization captures the notion that many differently-worded questions are reducible to identical underlying semantics (captured by a program), making the program space, in a sense, a compression of the question space. As mentioned in Section 1 the model also captures the independence of questions from answers for an image given programs, which is also an intuitive desiderata for interpretable neural-symbolic models[2].

Given the proposed model shown in Figure 2, we must address two tasks to put the model to use for VQA – learning model parameters, and performing inference for a specific query.

**Learning.** We will always condition on images $\mathbf{i}$ but may not observe underlying programs $\mathbf{z}$, so our per-data point training objective is the log-marginal-conditional-likelihood: $\log p(\mathbf{x}, \mathbf{a} \mid \mathbf{i})$. Since computing this objective involves an intractable summation over latent programs ($\log \sum_\mathbf{z} p(\mathbf{x}_j, \mathbf{z}, \mathbf{a}_j \mid \mathbf{i}_j)$), we use a variational approximation (Jordan et al., 1999). Specifically, let

- $q_\phi(\mathbf{z} \mid \mathbf{x})$ denote an amortized inference neural network parameterized by $\phi$, which instantiates a sequence to sequence LSTM (Hochreiter & Schmidhuber, 1997) model
- $p_{\theta_\mathbf{z}}(\mathbf{a} \mid \mathbf{z}, \mathbf{i})$ be a NMN parametrized by $\theta_\mathbf{z}$ run on image $\mathbf{i}$ with program $\mathbf{z}$,
- $p_\theta(\mathbf{x} \mid \mathbf{z})$ be a recurrent network (sequence to sequence model) parametrized by $\theta$ that maps programs to questions, and
- $p(\mathbf{z})$ be a prior over programs learned from data or based on program syntax directly , parameterized as an LSTM (Hochreiter & Schmidhuber, 1997) sequence model

Then, for datapoints without program supervision, the objective is:

$$\text{elbo}(\mathbf{a}, \mathbf{x}|\mathbf{i}) = E_{z \sim q_\phi(\mathbf{z}|\mathbf{x})}[\log p_{\theta_\mathbf{z}}(\mathbf{a}|\mathbf{z}, \mathbf{i}) + \log p_\theta(\mathbf{x}|\mathbf{z})] - \beta \text{KL}(q_\phi(\mathbf{z}|\mathbf{x}), p(\mathbf{z}))$$

It is easy to show that $\text{elbo}(\mathbf{x}, \mathbf{a}|\mathbf{i}) \leq \log p(\mathbf{x}, \mathbf{a}|\mathbf{i})$ for $\beta \geq 1$; as such, maximizing this objective is a form of approximate maximum likelihood training for distribution parameters $\theta_\mathbf{z}$, $\theta$, and $\phi$.

Additionally, for data points with annotations, we add the likelihood of the program given the question $\log r_\phi(\mathbf{z}|\mathbf{x})$ to the objective. Note that $r$ shares parameters $\phi$ with the inference network $q$. So our overall objective becomes:

$$\text{elbo}(\mathbf{a}, \mathbf{x}|\mathbf{i}) + \alpha \log r(\mathbf{z}|\mathbf{x}) \cdot [\![\mathbf{x} \in \mathcal{S}]\!] \tag{1}$$

where the indicator function $[\![\mathbf{x} \in \mathcal{S}]\!]$ is 1 if $\mathbf{x}$ is in the supervised set and 0 otherwise and $\alpha$ is a

---

[2]This view also matches finding from Hu et al. (2017) who find models linking questions to answer (via attention) operating in low-program supervision regimes often have syntactically poor programs despite leading to correct answers – i.e. the program need not be a sufficient statistic anymore to answer the question.

positive scaling factor. The goal is to learn the desired association of questions with programs from the second term, and to propagate them to other questions using the first term.

**Inference.** At test time, we can answer questions for images by computing $p(\mathbf{a}|\mathbf{i}, \mathbf{x}) = \sum_z p(\mathbf{a}|\mathbf{i}, \mathbf{z})p(\mathbf{z}|\mathbf{x})$ using an approximation $p_q(\mathbf{a}|\mathbf{i}, \mathbf{x}) \approx E_{\mathbf{z}_i \sim q(\mathbf{z}|\mathbf{x})}p(\mathbf{a}|\mathbf{i}, \mathbf{z}_i)$.

**Challenges.** While the elbo term in Equation (1) appears similar to a standard ELBO for joint latent variable models (Vedantam et al., 2018; Wang et al., 2016b; Suzuki et al., 2017), there are a few important departures:

- Firstly, the latent variable $\mathbf{z}$ is not a Gaussian or categorical random variable (which admit straightforward reparameterizations (Maddison et al., 2017; Jang et al., 2016)) but rather a sequence of discrete tokens, which complicates the optimization problem in Equation (1).
- Secondly, the conditional log-likelihood for $\mathbf{a}$ is parameterized by a NMN that is dynamically assembled based on sample programs drawn from $q(\mathbf{z}|\mathbf{x})$. Depending upon the uncertainty of the program posterior, program tokens may not be consistently grounded across examples, making learning the appropriate computation for the corresponding neural modules difficult.
- Lastly, our observations $\mathbf{x}$ are explained by powerful "autoregressive" sequence models. Learning meaningful latent variable models that are informative of observations for such models is difficult (c.f. (Bowman et al., 2016; Chen et al., 2016; Alemi et al., 2018)) as powerful decoder models can explain the data using stochasticity from sampling (referred to as autodecoding in Alemi et al. (2018)). Consequently, it also becomes hard to learn meaningful NMN parameters from the previous step.

In the following sections, we present a training regime and adjustments to the objective in Equation (1) that make optimization more tractable and enable training NMNs with significantly reduced question-aligned program supervision. While instantiated for question answering, it is not difficult to conceive versions of this regime for other problems relevant to probabilistic neural-symbolic models.

## 2.1 Optimizing the ELBO for V-NMN

Rather than approaching the joint objective directly, we break ELBO training into three stages:

1. **Question coding**, where we learn a latent 'code' for questions in the latent space of programs,
2. **Module training**, where we learn the 'execution' of the programs from the first stage of training, grounding tokens in programs into the operations they perform on images; and
3. **Joint training**, where we train both stages jointly given the 'warm-starts' from the other phases.

**Question Coding.** The goal of this stage is to learn a code for questions in the latent space of programs – *i.e.* an informative mapping from questions to programs. Given some set of question-aligned programs, we find this coding process effectively propagates these supervised groundings to unsupervised questions, which enable interpretable explanations / groundings of queries to explanations of "what" the model is doing, which is our primary goal.

During question coding, we drop the $p_{\theta_z}(\mathbf{a}|\mathbf{z}, \mathbf{i})$ term from Eq. 1, yielding the objective

$$J(\mathbf{x}; \phi, \theta) = E_{z \sim q_\phi(\mathbf{z}|\mathbf{x})}[\log p_\theta(\mathbf{x}|\mathbf{z})] - \beta \mathrm{KL}(q_\phi(\mathbf{z}|\mathbf{x}), p(\mathbf{z})) + \alpha \log r_\phi(\mathbf{z}|\mathbf{x}) \cdot [\![\mathbf{x} \in \mathcal{S}]\!] \quad (2)$$

Recent theory from Alemi et al. (2018) prescribes changes to the ELBO, setting $\beta < 1$ as a recipe to learn representations which avoid learning degenerate latent representations in the presence of powerful decoders. Essentially, Alemi et al. (2018) identify that upto a constant factor, the negative log-marignal likelihood term $D = -E_{z \sim q_\phi(\mathbf{z}|\mathbf{x})}[\log p_\theta(\mathbf{x}|\mathbf{z})]$, and the kl diveregence term $R = \mathrm{KL}(q_\phi(\mathbf{z}|\mathbf{x}), p(\mathbf{z}))$ bound the mutual information between the data $\mathbf{x}$ and the latent variable $\mathbf{z}$ as follows:

$$H - D \leq I(\mathbf{x}, \mathbf{z}) \leq R \quad (3)$$

where $H$ is the entropy of the data, which is a constant. One can immediately notice that the standard elbo $= -D - R$. This means that for the same value of the elbo one can get different models which make drastically different usage of the latent variable (based on the achieved values of $D$ and $R$). Thus, one way to achieve a desired behavior (of say high mutual information $I(\mathbf{x}, \mathbf{z})$), is to set $\beta$ to a lower value than the standard elbo (c.f. Eqn. 6 in Alemi et al. (2018).) This aligns with our goal in question coding, and so we use $\beta < 1$. As this no longer a valid variational lower bound, denote our objective as $J(\mathbf{x}; \phi, \theta)$ from now on.

We optimize the above objective using the score function estimator (Glynn, 1990), also referred to as REINFORCE (Williams, 1992) applied to the first term, with a moving average baseline. Finally,

while the KL term (in the reward) encourages the model to generate high-likelihood plausible sequences, we also penalize the model for producing syntactically invalid programs.

Although gradients for the KL term in Equation (2) can be computed by decomposing the KL for the sequence into per-timestep monte-carlo gradients, we found that having one term with per-timestep gradients and another that only receives sequence-level rewards (the likelihood term) skewed the model towards minimizing the KL, leading to auto-decoding solutions. Thus, we apply REINFORCE on the entire sequence and compute rewards using the full monte carlo gradient, *i.e.*, we do not optimize the KL term separately, but instead maximize:

$$E_{\mathbf{z} \sim q_\phi(\mathbf{z}|\hat{\mathbf{x}})} \left[ \log \left[ \frac{p_\theta(\hat{\mathbf{x}}|\mathbf{z})p(\mathbf{z})^\beta}{q_\phi(\mathbf{z}|\hat{\mathbf{x}})^\beta} \right] \right] + \alpha \log r_\phi(\hat{\mathbf{z}}|\hat{\mathbf{x}}) \cdot [\![\mathbf{x} \in \mathcal{S}]\!] \tag{4}$$

where $\hat{\mathbf{x}}, \hat{\mathbf{z}}$ are the questions and programs we observe in the dataset.

For comparison, the semi-supervised approach in Johnson et al. (2017) can be seen as optimizing the last term on supervised training data, while our question coding stage optimizes the elbo in addition to the last term. Our results in Section 5 show that the proposed question coding formulation significantly improves performance relative to Johnson et al. (2017) in terms of explaining "what" the model is doing on test programs.

**Module Training.** Having obtained a code for questions in the latent space, we train the parameters of the NMN for question answering. The network modules are trained to perform their corresponding tasks. For example, if the question coding learned to associate the program token $v$ with the question phrase 'to the left', then the goal for this second stage is to train the module corresponding to $v$ to actually perform this operation (*e.g.* `transform[left]`) when executed on images.

We do this by optimizing the joint $\mathrm{elbo}(\mathbf{x}, \mathbf{a}|\mathbf{i})$ in (1) with respect to the parameters $\theta_{\mathbf{z}}$ of the neural modules. We write this objective for each instance as

$$J(\mathbf{a} \mid \mathbf{x}, \mathbf{i}; \theta_{\mathbf{z}}) = E_{z \sim q(\mathbf{z}|\mathbf{x})}[\log p_{\theta_z}(\mathbf{a}|\mathbf{i}, \mathbf{z})] \tag{5}$$

Notably, while we write this objective as a function of all neural modular network parameters $\theta_{\mathbf{z}}$, for a given instance, only those modules present in the program $z \sim q(\mathbf{z}|\mathbf{x})$ are updated. In practice, we take approx. argmax decoded programs from $q_\phi(\mathbf{z}|\mathbf{x})$ to simplify training, but perform sampling during joint training in the next stage. To be more specific, we run beam search (Vijayakumar et al., 2018) with a beam size of 1 to do this. Intuitively, we think of this as a bias *vs.* variance tradeoff, doing argmax enables us to trade off bias for high-variance from a posterior which (in the pathological case) could match the prior from the first stage of training. This is also consistent with standard practice for doing inference in sequence models, where beam search is the standard inference tool (and sampling often has issues due to teacher forcing based MLE training, which is how we do semi-supervised learning) (Bengio et al., 2015).

**Joint Training.** Having trained the question code and the neural module network parameters, we tune all terms jointly, with the following objective, which adds scaling factors $\alpha$, $\beta$ and $\gamma$, and an extra semi-supervised learning term relative to Equation (1):

$$J(\mathbf{x}, \mathbf{a}|\mathbf{i}) = E_{z \sim q_\phi(\mathbf{z}|\mathbf{x})}[\gamma \log p_{\theta_{\mathbf{z}}}(\mathbf{a}|\mathbf{z}, \mathbf{i}) + \log p_\theta(\mathbf{x}|\mathbf{z})] - \beta \mathrm{KL}(q_\phi(\mathbf{z}|\mathbf{x}), p(\mathbf{z})) + \alpha \log r_\phi(\mathbf{z}|\mathbf{x}) \tag{6}$$

The gamma factor $\gamma$ controls the scaling of the likelihood term for the answers relative to the question (which has more bits of information), similar scaling factors have been for joint models in Vedantam et al. (2018). Similar to Equation (4), we optimize the full monte carlo gradient for the above objective as well. The full training procedure is shown in Algorithm 1 below.

## 3 RELATED WORK

We first explain other related work in neuro-symbolic models, the key differences between our formulation and closely related NMN work (Johnson et al., 2017; Hu et al., 2017; 2018), and then connect our work to other themes and directions of research in literature.

**Neural-symbolic models** Valiant (2003) identifies neuro-symbolic models, which are able to reason as well as learn efficiently as one of the key problems in computer science. A rich literature of work exists in this area, around two major threads: firstly, there is a line of work in building hybrid models

---

**Algorithm 1** V-NMN Training

---

1: Given: $p(\mathbf{z})$, Initialize $q_\phi(\mathbf{z}|\mathbf{x})$, $p_{\theta_z}(\mathbf{a}|\mathbf{z},\mathbf{i})$, $p_\theta(\mathbf{x}|\mathbf{z})$, moving average baseline $B = 0$, $\beta = 0.1, \alpha = 100.0, \delta = 0.99$.

2: **Question Coding:** Given $\hat{\mathbf{x}}, \hat{\mathbf{z}} \sim p_{data}(\mathbf{x}, \mathbf{z})$, sample $\mathbf{z_i} \sim q_\phi(\mathbf{z}|\hat{\mathbf{x}})$ [Repeat until convergence]

3: $\quad \nabla_\phi \leftarrow (R - B)\log q_\phi(\mathbf{z}_i|\hat{\mathbf{x}}) + \alpha\nabla_\phi \log r_\phi(\hat{\mathbf{z}}|\hat{\mathbf{x}})$; where $R = \log\left[\frac{p_\theta(\hat{\mathbf{x}}|\mathbf{z}_i)p(\mathbf{z}_i)^\beta}{q_\phi(\mathbf{z}_i|\hat{\mathbf{x}})^\beta}\right]$

4: $\quad \nabla_\theta \leftarrow \nabla_\theta \log p_\theta(\hat{\mathbf{x}}|\mathbf{z}_i)$

5: $\quad B \leftarrow B + (1 - \delta)(R - B); \nabla_\theta \leftarrow \nabla_\theta \log p(\hat{\mathbf{x}}|\mathbf{z}_i)$

6: **Question Coding:** Given $\hat{\mathbf{x}}, \hat{\mathbf{z}} \sim p_{data}(\mathbf{x}, \mathbf{z})$, sample $\mathbf{z_i} \sim q_\phi(\mathbf{z}|\hat{\mathbf{x}})$ [Repeat until convergence]

7: **Module Training:** Given $\hat{\mathbf{x}}, \hat{\mathbf{a}}, \hat{\mathbf{i}} \sim p_{data}(\mathbf{x}, \mathbf{a}, \mathbf{i})$, $\mathbf{z}_i = \text{argmax } q(\mathbf{z}|\hat{\mathbf{x}})$ [Repeat until convergence]

8: $\quad \nabla_{\theta_z} \leftarrow \nabla_{\theta_z} \log p(\hat{\mathbf{a}}|\mathbf{z}_i, \hat{\mathbf{i}})$

9: **Joint Training:** Given $\hat{\mathbf{x}}, \hat{\mathbf{a}}, \hat{\mathbf{i}}, \hat{\mathbf{z}} \sim p_{data}(\mathbf{x}, \mathbf{a}, \mathbf{i}, \mathbf{z})$, $\mathbf{z}_i \sim q(\mathbf{z}|\hat{\mathbf{x}})$, $\gamma$; Initialize $B = 0$, [Repeat until convergence]

10: $\quad \nabla_\phi \leftarrow (R - B)\nabla_\phi \log q_\phi(\mathbf{z}_i|\hat{\mathbf{x}}) + \alpha\nabla_\phi \log r_\phi(\hat{\mathbf{z}}|\hat{\mathbf{x}})$; where $R = \log\left[\frac{p_\theta(\hat{\mathbf{x}}|\mathbf{z}_i)p(\hat{\mathbf{a}}|\mathbf{z}_i,\hat{\mathbf{i}})^\gamma p(\mathbf{z}_i)^\beta}{q_\phi(\mathbf{z}_i|\hat{\mathbf{x}})^\beta}\right]$

11: $\quad B \leftarrow B + (1 - \delta)(R - B); \nabla_\theta \leftarrow \nabla_\theta \log p(\hat{\mathbf{x}}|\mathbf{z}_i); \nabla_{\theta_z} \leftarrow \nabla_{\theta_z} \log p(\hat{\mathbf{a}}|\mathbf{z}_i, \hat{\mathbf{i}})$

---

which are hybrids of symbolic reasoning and neural networks (Bader & Hitzler, 2005; Hölldobler & Kalinke), the second thread looks at studying if neural networks can indeed do symbolic reasoning and building models and architectures to achieve this for specific cases (Evans et al., 2018a; Rocktäschel et al., 2015). Our work is distinct in terms of being motivated by interpretability in the low-sample regime rather than getting connectionist/neural networks to explicitly do symbolic manipulations or building hybrid models that do logic.

More recently, Yin et al. (2018), propose a probabilistic latent variable model of structured programs which explain textual observations for tasks like parsing, *i.e.*, the model from Yin et al. (2018) does not capture "how" to execute the programs that it generates. Our model is more general in the sense that we have a second modality which is the output of what gets executed when the program runs, and we model both jointly in our model, by learning how to execute the program. Work from Yi et al. (2018) also shares the same goal of doing question answering with limited question program supervision while preserving the semantics of human programs. While they to approach this problem by simplifying the $p(\mathbf{a}|\mathbf{i}, \mathbf{z})$ mapping (in context of our model) by first converting the image into a table, we approach this goal by modeling a stochastic latent space. Thus the two approaches are orthogonal, as one can use our probabilistic latent space in conjunction with their model as well to better share statistics between question-program pairs. Overall, compared to these works, our work addresses probabilistic neural-symbolic learning in a more general setting, by learning how to execute programs (on raw-pixels), as well as handling parsing of questions in an end to end manner.

**Connection to prior work on NMNs.** Our V-NMN model is inspired in its parameterization of program to answer mapping from prior work on visual question answering using neural module networks such as Johnson et al. (2017); Hu et al. (2017; 2018). Distinct from these previous approaches, our explicit goal is to preserve the semantics of how a human specifies a task to be broken down in terms of instructions, while previous work was motivated by building inductive biases for reasoning (which turn out to be useful for our model as well). Given this goal, we construct a latent variable model that in some sense embeds previous work. The works from Hu et al. (2017; 2018) are more distant from our work, as they use continuous attention modules, which do not capture independence assumptions of the program being a sufficient statistic for answering the question (which makes them harder to specify completely for a human, as they would have to hallucinate attention parameters). However, apart from our focus on preserving the lexicon/specification given by a human, both approaches share the same high level goals of providing interpretable question answering with minimal supervision.

**Beyond Gaussian Latents in Amortized Variational Inference.** There has been exciting progress in developing lower-variance estimators for discrete generative models (Tucker et al., 2017; Mnih & Gregor, 2014; Titsias & Lázaro-Gredilla, 2015; Gu & Ling, 2015). In particular, Maddison et al. (2017); Jang et al. (2017) propose the use of the concrete distribution on the probability simplex for providing biased but low variance estimators for discrete latent variable models by considering a reparametrization of samples (which asymptotically approach those from a categorical distribution). The contribution in our work is orthogonal to these approaches, how one can apply

such parameterizations to neuro-symbolic models. More relevant to our work is Miao & Blunsom (2016) which proposes a discrete sequential latent variable model for sentence summarization, with summaries as the latent variable. In contrast to their model, ours is a joint model where the latent space consists of programs, not summaries, which have to explain questions and answers to images.

**Joint Variational Autoencoder Models.** Vedantam et al. (2018) propose a joint model for images and attributes for "visual imagination", Suzuki et al. (2017) propose a closely related model and show that one can use the model for conditional generation, while Wang et al. (2016a) propose a deep variational canonical correlation analysis model which learns a joint space between two modalities for denoising and representation learning. In contrast to these approaches, our joint model assumes a discrete structured latent variable (instead of a Gaussian), does semi-supervised training, which is possible for us, but not possible when there are no observations from the latent available for supervision, and is targeted towards neural module networks and question answering.

**Program Induction and Neural Architecture Search.** In program induction one is typically interested in learning to write programs given specifications of input-output pairs (Reed & de Freitas, 2015; Kalyan et al., 2018), and optionally language (Neelakantan et al., 2015; Guu et al., 2017) or vision (Gaunt et al., 2017). The key difference between program induction and our problem is that program induction assumes the tokens/instructions in the language are grounded to the executor, *i.e.*, it is assumed that once a valid program is generated there is a black-box execution engine that can execute it, while we learn the execution engine from scratch, and learn the grounding from program symbols to parameters of a neural network for execution. Neural architecture search (Zoph & Le, 2016) is another closely related problem where one is given input output examples for a machine learning task, and the goal is to find a "neural program" that does well on the task. In contrast, ours can be seen as the problem of inducing programs *conditioned* on particular inputs (questions), where the number of samples seen per question is significantly lower than the number of samples one would observe for a machine learning task of interest.

## 4 EXPERIMENTS

**Dataset.** We perform our experiments on the SHAPES dataset (Andreas et al., 2016a). This dataset has been used in previous works (Andreas et al., 2016a; Hu et al., 2017) on visual question answering using neural module networks, and is intended to be a test for compositional generalization. The dataset has 244 unique binary questions (*i.e.* having yes/no answers) and 15,616 images in total (Andreas et al., 2016a). The dataset also has annotated programs for each of the questions. We use train, val, and test splits of 13,568, 1,024, and 1,024 ($\mathbf{x}, \mathbf{z}, \mathbf{i}, \mathbf{a}$) triplets respectively. No questions are repeated across these splits. Further, each question is asked for 64 different images, ensuring that a question can be answered correctly across all these instances only if it is processing the image in a reasonable way (as dictated by the underlying program), as opposed to leveraging any image-question correlations or biases that are unlikely to generalize across images.

Thus, the dataset explicitly tests for compositional generalization and "reasoning" in the task of visual question answering. The longest questions in the dataset are of length 11 and shortest are of length 4, while the longest programs are of length 6 and shortest programs are of length 4. The size of the question vocabulary $\mathcal{V}_{\mathbf{x}}$ is 14 and the program vocabulary $\mathcal{V}_{\mathbf{z}}$ is 12.

**Training.** To simulate a data-sparse regime, we restrict the set of question-aligned programs to 5, 10, 15, or 20% of unique questions – such that even at the highest level of supervision, programs for 80% of unique question have never been seen during training. Unless specified explicitly, we train our program prior using a set of 1848 unique programs simulated from the syntax (more details of this can be seen in the appendix). In general, it might not be possible to exhaustively enumerate the space of all valid strings to learn the prior. Thus, we also evaluate the effect of training the prior with varying-sized subsets of the 212 programs present in the training set (without using any corresponding questions or answers) to check how robust overall performance is to the choice of the prior (in the appendix).

We find sequence-level policy-gradient based training in this setting can be highly variable – with some runs failing spectacularly while others succeed – for both our method and to some extent NMN (Johnson et al., 2017). For fairness and to reduce the impact of stochasticity, we perform the following stage-wise training regime for all models. First, the question-coding stage is run for 5 different

random seeds, producing multiple program predictors. Programs taken from the model that performs the best on the validation set are used to train 10 runs of the module training stage. Finally, the best of these module networks (based on val) is trained jointly with the program predictor under settings of $\gamma \in \{1, 10, 100\}$ and the max (on val) is chosen. We report mean and variance accuracies on test from this training regime over four random sets of supervisory programs. Our evaluation metrics are as follows: a) question-coding uses reconstruction accuracy (of getting the entire question right) and program prediction accuracy (of matching the ground truth programs) on the *training* set, since the second stage, module training will use inferences from the first stage (on the training set to train modules), b) module training and joint training: visual question answering accuracy, by counting matches to ground truth answers.

**Implementation Details**: Following Johnson et al. (2017) and Hu et al. (2017), we treat the program $\mathbf{z} = \{z_1, \cdots, z_K\}$ as a variable length sequence of tokens with an end of sequence token, which corresponds to the prefix traversal of an execution tree. Given this representational choice, we learn the prior $p(\mathbf{z})$ using maximum likelihood training on simulated samples from known syntax of programs. An alternative, interesting approach to try would be to use something like Janz et al. (2017), who acquire a generative model more directly from validity annotations. The $\log p(\mathbf{a}|\mathbf{z}, \mathbf{i})$ term in Equation (1) is parameterized by an NMN that is dynamically assembled given the program string, while the generative model $p(\mathbf{x}|\mathbf{z})$, and inference ($q(\mathbf{z}|\mathbf{x})$) terms are both sequence to sequence models. Our sequence to sequence models are based on LSTM (Hochreiter & Schmidhuber, 1997) cells. For the NMN programs, we use syntax similar to that used by Hu et al. (2017), but we replace their question-attention neural modules with "regular" modules. For clear comparison we adapt the objectives from the baseline approach, Johnson et al. (2017), to our representational choices. More details can be found in appendix.

## 5 RESULTS

We focus on evaluating approaches by varying the fraction of question-aligned programs (which we denote $\%\mathbf{x} \leftrightarrow \mathbf{z}$) in Table 1. To put the numbers in context, a baseline LSTM + image model, which does not use module networks, gets an accuracy of 63.0% on Test (see Andreas et al. (2016a); Table. 2). This indicates that SHAPES has highly compositional questions which is challenging to model via conventional methods for Visual Question Answering (Agrawal et al., 2015). Overall, we make the following observations:

- **Our V-NMN approach consistently improves performance in data-sparse regimes.** While both methods tend to improve with greater program supervision, V-NMN quickly outpaces NMN (Johnson et al., 2017), achieving test accuracies over 20-35% points higher for $>5\%$ program supervision. Notably, both methods perform similarly poorly on the test set given only 5% program supervision, suggesting this may be too few examples to learn compositional reasoning.
- **Our question coding stage greatly improves initial program prediction.** Our V-NMN approach to question coding gets to approximately double the question coding (reconstruction) accuracy as the NMN approach (col 1, question coding). This means that it effectively propagates groundings from question-aligned programs during the coding phase. Consequently, we also see improved VQA performance after the module and joint training stages which are based on predicted programs.
- **Question reconstruction is well-correlated with overall accuracy.** Despite the program space being trained directly as a latent code for questions, we find question reconstruction accuracy rises with the amount of supervision – confirming that inference in a space of discrete structured latent variables is difficult to do without some supervision to guide the mapping.

Figure 3 shows sample programs for each model. We see the limited supervision negatively affects NMN program prediction, with the 5% model resorting to simple `Find[X]`→`Answer` structures. Interestingly, we find that the mistakes made by the V-NMN model, *e.g.*, green in 5% supervision (top-right) are also made when reconstructing the question (also substituting green for blue). Further, when the token does get corrected to blue, the question also eventually gets reconstructed (partially correctly (10%) and then fully (15%)), and the program produces the correct answer. This indicates that there is high fidelity between the learnt question space, the answer space and the latent space.

**Effect of the Program Prior $\mathbf{p}(\mathbf{z})$.** Next, we explore the impact of regularizing the program posterior to be close to the prior $p(\mathbf{z})$, for different choices of the prior. Firstly, we disable the KL-divergence term by setting $\beta = 0$ in Equation (2) recovering a deterministic version of our model. Compared to our full model at 10% supervision, the performance on question coding drops to 12.7% program

| | $\%\mathbf{x} \leftrightarrow \mathbf{z}$ | Validation During Training Stages | | | | Test Accuracy |
|---|---|---|---|---|---|---|
| | | [I] Question Coding | | [II] Module Training | [III] Joint Training | |
| | | (Reconstruction) | (Program Prediction) | (VQA Accuracy) | (VQA Accuracy) | (VQA Accuracy) |
| NMN (Johnson et al., 2017) | 5 | - | $9.28_{\pm 1.91}$ | $61.56_{\pm 3.59}$ | $63.08_{\pm 0.78}$ | $60.06_{\pm 3.88}$ |
| V-NMN (Ours) | | $54.86_{\pm 6.75}$ | $17.90_{\pm 3.48}$ | $63.21_{\pm 3.12}$ | $66.61_{\pm 1.58}$ | $61.40_{\pm 4.41}$ |
| NMN (Johnson et al., 2017) | 10 | - | $24.30_{\pm 2.39}$ | $60.31_{\pm 3.37}$ | $66.51_{\pm 4.10}$ | $61.99_{\pm 0.96}$ |
| V-NMN (Ours) | | $87.52_{\pm 6.45}$ | $65.45_{\pm 11.88}$ | $81.34_{\pm 8.61}$ | $92.85_{\pm 4.89}$ | $88.01_{\pm 6.45}$ |
| NMN (Johnson et al., 2017) | 15 | - | $47.67_{\pm 5.02}$ | $69.47_{\pm 9.87}$ | $62.43_{\pm 0.49}$ | $61.32_{\pm 2.36}$ |
| V-NMN (Ours) | | $99.66_{\pm 0.19}$ | $86.23_{\pm 3.29}$ | $93.24_{\pm 3.17}$ | $97.97_{\pm 1.20}$ | $94.63_{\pm 2.65}$ |
| NMN (Johnson et al., 2017) | 20 | - | $58.37_{\pm 3.30}$ | $66.17_{\pm 7.02}$ | $80.68_{\pm 18.00}$ | $78.59_{\pm 19.27}$ |
| V-NMN (Ours) | | $99.88_{\pm 0.20}$ | $90.95_{\pm 0.90}$ | $91.12_{\pm 2.67}$ | $99.13_{\pm 0.72}$ | $95.63_{\pm 1.31}$ |

Table 1: Results in %, on the SHAPES dataset with varying amounts of question-program supervision ($\%\mathbf{x} \leftrightarrow \mathbf{z}$), $\beta = 0.1$.

prediction accuracy (from 65.45%), while the performance on question reconstruction improves from 87.52% to 93.8%. Without the KL term, the model focuses solely on reconstruction and fails to learn compositionality in the latent $\mathbf{z}$ space, such that supervised grounding are poorly propagated to unsupervised questions as evidenced by the drop in program prediction accuracy. Note that this ties well to our high-level goal to better leverage the semantics of the lexicon provided by the user for interpretable question answering.

**Beam search *vs*. sampling for module training.** As explained in Section 2, we use beam search instead of sampling to optimize Equation (5). In this section we empirically study this design choice. We generally find that in low-supervision settings, where the question coding stage might not produce correct programs (for questions) as frequently, using sampling form the sequence model $q(\mathbf{z}|\mathbf{x})$ instead of beam search leads to a drop in module training performance. For example, with 10% supervision, the module training accuracy drops from $81.34 \pm 8.61$ (with beam search) to $76.31 \pm 8.41$ (with sampling). With more supervision the choice of beam search or sampling does not matter, which makes intuitive sense, as the this means that the programs that get sampled tend to be the correct ones more often (that is, the confidence of the model increases with the improvement in correctness).

**Effect of optimizing the true ELBO, $\beta = 1$.** Next, we empirically validate the claim that approximate inference learned using the ELBO leads to autodecoding behavior as predicted by Alemi et al. (2018), and that better program representations for our problem can be found for $\beta < 1$ (as shown above). At 15% program supervision, VQA accuracy drops from 96.90% ($\pm 0.96$) at $\beta = 0.1$ to 78.49% ($\pm 1.37$) at $\beta = 1$ (for the empirical prior). Question reconstruction drops from 97.43 % ($\pm 3.42$) at $\beta = 0.1$, to 43.40 % ($\pm 5.34$) at $\beta = 1$, showing auto-decoding behavior (as predicted by theory).

Finally, the N2NMN approach (Hu et al., 2017) evaluates their question-attention based module networks in the fully unsupervised setting, getting to 96.19% on TEST. However, the programs in this case become non-compositional (see Section 3), as the model leaks information from questions to answers via attention, meaning that programs no longer carry the burden of explaining the observed answers. This makes the modules less interpretable; which is one of the core motivations of this neural-symbolic class of module networks in the first place.

## 6 DISCUSSION AND CONCLUSION

In this paper we presented a novel, probabilistic neural symbolic model for interpretable visual question answering, that provides explanations of "what" the model is doing on unseen questions given a minimal number of annotated symbolic traces or programs of "what it should do". We demonstrate that our formulation provides more interpretable explanations than previous work on visual question answering called neural module networks, on dataset of compositional questions about shapes (Andreas et al., 2016a).

The key to our approach is a model with programs as a stochastic latent variable, which leads to better sharing of statistics across questions, yielding a latent space where program annotations for known questions propagate effectively to unknown / novel questions without annotations.

In general, the model family we study is quite rich, and also supports other inference queries such as counter factual explanations of what programs could have led to particular answers for a given image.

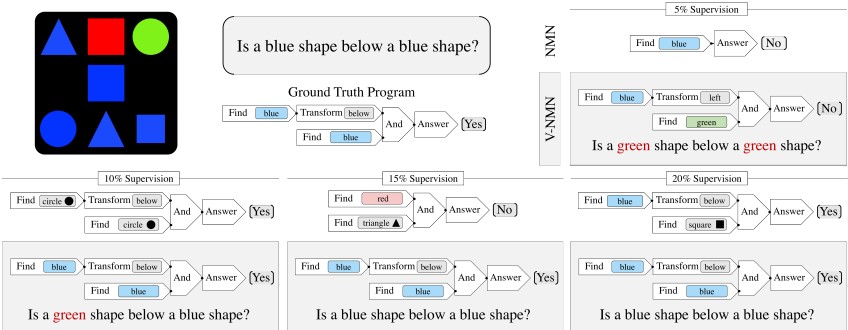

Figure 3: Qualitative results: Image with the corresponding question, ground truth program, and ground truth answer (top left). Predicted programs and answers of different models (NMN in white, V-NMN in gray) and resconstructed question (V-NMN) for variable amount of supervision. We find that V-NMN finds the right answer as well as the program more often than NMN (Johnson et al., 2017).

We hope our work inspires more work on applying such ideas to other tasks requiring compositional reasoning such as planning, navigation etc.

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

APPENDIX

### .1 IMPLEMENTATION DETAILS

We provide more details on the modeling choices and hyperparameters used to optimize the models in the main paper.

**Sequence to Sequence Models:** All our sequence to sequence models described in the main paper (Section. 2) are based on LSTM cells with a hidden state of 128 units, single layer depth, and have a word embedding of 32 dimensions for both the question as well as the program vocabulary.

The when sampling from a model, we sample till the maximum sequence length of 15 for questions and 7 for programs.

**Image CNN:** The SHAPES images are of `30x30` in size and are processed by a two layered convolutional neural network, which in its first layer has a `10x10` filters applied at a stride of 10, and in the second layer, it does `1x1` convolution applied at a stride of 1. Channel widths for both the layers are 64 dimensional.

**Moving average baseline** For a reward $R$, we use an action independent baseline $b$ to reduce variance, which tracks the moving average of the rewards seen so far during training. Concretely, the form for the $q(\mathbf{z}|\mathbf{x})$ network is:

$$\nabla J(\theta) = \mathbf{E}_{\mathbf{z} \sim q(\mathbf{z}|\mathbf{x})}[(R - b_t)\nabla \log q(\mathbf{z}|\mathbf{x})] \tag{7}$$

where, given $D$ as the decay rate for the baseline,

$$b_t = b_{t-1} + D * (R - b_{t-1}) \tag{8}$$

is the update on the baseline performed at every step of training.

Note that in practice, when estimating the gradient for the variational lower bound, we consider the total derivative (which adds a term having a gradient with respect to $R$ above), since in the case of sample-based optimization of the elbo, the reward itself is also a function of the variational parameters, via. both the sample used estimate the reward, as well as the reward directly being a function of $q(\mathbf{z}|\mathbf{x})$.

**Hyperparameters for training:** We use the ADAM optimizer (Kingma & Ba, 2014) with a learning rate of `1e-3`, a minibatch size of 576, and use a moving average baseline for REINFORCE, with a decay factor of 0.99. Typical values of $\beta$ are set to 0.1, $\alpha$ is set to 100 and $\gamma$ is chosen on a validation set among (1.0, 10.0, 100.0).

**Simulating programs from known syntax:** We follow a simple two-stage heuristic procedure for generating a set of samples to train the program prior in our model. Firstly, we build a list of possible future tokens given a current token, and sample a random token from that list, and repeat the procedure for the new token to get a large set of possible valid sequences – we then pass the set of candidate sampled sequences through a second stage of filtering based on constraints from the work of Hu et al. (2017).

### .2 ADDITIONAL RESULTS

**Empirical *vs.* syntactic priors** While our default choice of the prior $p(\mathbf{z})$ is from programs simulated from the known syntax, it might not be possible to exhaustively enumerate all valid program strings in general. Here, we consider the special case where we train the prior on a set of *unaligned*, ground truth programs from the dataset. Interestingly, we find that the performance of the question coding stage, especially at reconstructing the questions improves significantly when we have the syntactic prior as opposed to the empirical prior (from $38.39 \pm 11.92\%$ to $54.86 \pm 6.75\%$ for 5% program supervision). In terms of program prediction accuracy, we also observe marginal improvements in the cases where we have 5% and 10% supervision respectively, from $56.23 \pm 2.81\%$ to $65.45 \pm 11.88\%$. When more supervision is available, regularizing with respect to the broader, syntactic prior hurts performance marginally, which makes sense, as one can just treat program supervision as a supervised learning problem in this setting.

