# OpenReview forum: "Probabilistic Neural-Symbolic Models for Interpretable Visual Question Answering"
_ICLR.cc/2019/Conference_

### Official Review · AnonReviewer3 · 2018-10-16
**A good piece of work**

**Rating:** 7
**Confidence:** 3

**Review:**

The paper presents a new approach for performing visual query answering. System responses are programs that can explain the truth value of the answer.
In the paper, both the problems of learning and inference are taken into account.
To answer queries, this system takes as input an image and a question, which is a set of word from a given vocabulary. Then the question is modeled by a plan (a series of operation that must be performed to answer the query).  Finally, the found answer with the plan are returned. To learn the parameters of the model, the examples are tuples composed by an image, a question, the answer, and the program.
Experiments performed on the SHAPES dataset show good performance compared to neural model networks by Johnson et al.

The paper is well written and clear. I have not found any specific problems in the paper, the quality is high and the approach seems to me to be new and worth studying.
The discussion on related work seems to be good, as well as the discussion on the results of the tests conducted.

On page 5, in equation (3) it seems to me that something is missing in J. Moreover, In Algorithm 1, in lines 4 and 9, the B after the arrow should be written in italic.

Overall, there are several typos that must be corrected. I suggest a double check of the English. For example:
- page 3, "as modeling *uncertaintly* should..."
- page 6, "Given this goal, we *consrtuct* a latent *varible* ..."
- page 8, in paragraph "Effect of optimizing the true ELBO", the word "that" is repeated twice in the 3rd row
- page 13, "for the" repeated twice in "Moving average baseline" paragraph. Also, in the last line of this paragraph, the sentence seems incomplete.



Pros
- The results are convincing
- The approach is clearly explained

Cons
- English must be checked

---

### Official Review · AnonReviewer2 · 2018-11-02
**Need improvement on the presentation**

**Rating:** 6
**Confidence:** 3

**Review:**

This paper proposes a variational neural module networks (V-NMN), which compared to neural module networks (NMNs), is formed in a probabilistic aspect. The authors compare the performance of V-NMN and NMN on SHAPES dataset.

I find the technical part is hard to follow. To optimize the objective function, it involves many challenges. The authors described those challenges as well. It is not clear to me how those challenges are solved in section 2.1. I think that the presentation in section 2.1 needs to provide more details.

In the experiment, the authors only compare their work with NMNs without comparing it with other approaches for visual question answering. Besides accuracy, does V-NMN provide new applications that NMNs and other VQA models is not applicable because of the probabilistic formulation?

---

> ### Comment · Area_Chair1 · 2018-11-07
> **Please consider other reviews**
>
> Reviewer 2's review is very short. I do not see much substance or argument supporting the quite strict score (4) in favour of rejecting the paper. Regardless of the fact that it does not harmonise with the assessments provided by the other reviewers, it is not appropriate to make a recommendation of this nature without giving the authors a clear indication of what needs improving in the paper. If the true failing of this paper is that the technical part is hard to follow, is this due to poor presentation, or due to concerns with the actual applicability or scalability of the method proposed?
>
> Please take a moment to read the other reviews, in particular Reviewer 3, as well as the author response(s), if and when they are made, and consider whether you can flesh out the concerns underlying your assessment in a way which the authors can respond to, rebut, or take into account when revising their paper.

---

> > ### Comment · AnonReviewer2 · 2018-11-07
> > **Provide more details based on AC's comments**
> >
> > I provide more details for my reviews based on AC's comments.
> >
> > First, I agree with the other two reviewers that the authors present the problem and their idea well and show V-NMN outperforms NMN in experiments. My concerns, however, are about the technical section 2.1, which I do not see other two reviewers comment on that.
> >
> > I think that section 2.1 is hard to follow because the presentation is vague and lack details. There are several places pointing the readers to check other references or briefly summarizing the ideas where I think technical details should be provided.
> >
> > Technically, I have concerns about how the authors solve the challenges they mentioned in section 2.1. In the question coding stage, the goal is to learn an informative mapping from questions to programs, which based on my understanding, is q_\phi(z|x). As the authors mention, the prior distribution for p(z) is not a Gaussian distribution. How do the authors design the prior distribution p(z) and the variational distribution q_\phi(z|x), and how do the authors optimize the KL(q_\phi(z|x), p(z)) term? I agree with the authors that it is not an easy question. The presentation from the authors for this hard question is vague and lack technical details. Based on my understanding, I also think the solution is heuristic.
> >
> > Second, in order to optimize q_\phi(z|x) in objective (1), the E_{z\sim q_\phi(z|x)}[\log p_{\phi_z}(a|z,i)] term has an effect as well. In the question coding stage, this term is totally removed. In order for the objective (1) to be a valid ELBO for log p(x, a|i), \beta need to be >=1. As the authors pointed out, they need to relax \beta to < 1 and it is no longer an ELBO. If the readers want to understand why it is good to relax \beta to < 1, they need to check Alemi et al. (2018), which is the only information the authors provide.
> >
> > In the module training stage, the goal is to learn the NMN for question answering with the objective (3) E_{z\sim q_\phi(z|x)}[\log p_{\phi_z}(a|z,i)]. After the authors obtain an estimate q_\phi(z|x) in question coding stage, it is possible to optimize this objective by sampling on q(z|x). Instead of using q(z|x), the authors presents "In practice, we take argmax decoded programs from q_\phi(z|x) to simplify training, but perform sampling during joint training in the next stage." Could the authors provide technical details for this "argmax decoded programs"?
> >
> > In the joint train stage, the authors have an objective (4), which is different from objective (1) with a scalar \gamma. The authors presents the reason for that is because it is similar as Vedantam et al. (2018), which again totally points readers to check other references when the relaxation seems to have issue. For objective (4), is \beta >=1 or < 1? Why the objective (1) is hard to train but objective (4) is possible to train? How do the question coding stage and module training stage helps to solve the challenges for objective (1) so that objective (4) is easier to train?
> >
> > Overall, my points for the technical section 2.1 is that at least the presentation needs to improve with more technical details so that readers can see how the authors solved the challenges they proposed. Also, I think the solution proposed by the authors is a heuristic way which is not clear how it solves those challenges. For other reviewers, do you think that section 2.1 need to provide more technical details as well?
> >
> > For a heuristic method, if the authors show promising experimental results, I value this kind of work as well. My concerns are that the literature review is focused on restricted related works without a comprehensive introduction of VQA works, for example,
> >
> > Interpretable Visual Question Answering by Visual Grounding from Attention Supervision Mining (Zhang et al. 2018)
> > Neural-Symbolic VQA: Disentangling Reasoning from Vision and Language Understanding (Yi et al. 2018).
> >
> > The authors only compares V-NMN with NMN in the experiments. Could the authors answer why it is enough to only compare with NMN without comparing other VQA methods? Besides accuracy improvement, is there any other benefit by using V-NMN compared to NMN?
> >
> > I am open to feedbacks and I will update my score if the authors can handle my concerns.

---

> > > ### Comment · Area_Chair1 · 2018-11-08
> > > **Thank you**
> > >
> > > Thanks for providing an expanded review so quickly. I'm sure the authors will find this helpful for the purpose of this discussion period.

---

> > > ### Comment · AnonReviewer3 · 2018-11-08
> > > **As regards Section 2.1**
> > >
> > > I have found section 2.1 good enough though it is more descriptive than mathematically detailed. However, comments of Rev #2 are right so I can agree with them, maybe authors could add such details in the appendix or in the remaining half page depending on how long would be the discussion.
> > > I still think that the document is worthy in this version. If the authors manage to add a good response to the Rev #2 comments, my current score will at least be confirmed if not increased.

---

> > > ### Author Response · Authors · 2018-11-27
> > > **Addressing R2's concerns [Part 1/2]**
> > >
> > > We reply to more specific concerns from R2 below, and hope to convince them that in light of the justifications (already in the paper and below) the proposed method is not heuristic and has sufficient clarity.
> > >
> > > ----------------------
> > > [R2] Besides accuracy improvement, is there any other benefit by using V-NMN compared to NMN?
> > >
> > > As mentioned in the paper, and pointed out by R1, V-NMN is not only more accurate but generates the right answers for the ``right’’ reasons by providing more correct program explanations for questions. This makes V-NMN more interpretable, which is one of our key stated goals. In general, interpretability is important for allowing humans to build trust to make actionable decisions from outputs generated by machines.
> > >
> > > ----------------------
> > > [R2] How do the authors design the prior distribution p(z) and the variational distribution q_\phi(z|x), and how do the authors optimize the KL(q_\phi(z|x), p(z)) term? [..]
> > >
> > > The appendix in the submitted version (Page. 14 in the revision) describes how we parameterize the variational distributions and the priors. Further, we have added text to the main paper (Page. 3, learning) to clarify how we parameterize prior and posterior distributions (LSTM recurrent neural network based sequence models), and highlight existing text in the paper (Page. 5, before Eqn. 4) and Algorithm 1, step 10 explaining how the KL divergence is optimized.
> > >
> > > ----------------------
> > > [R2]  If the readers want to understand why it is good to relax \beta to < 1, they need to check Alemi et al. (2018), which is the only information the authors provide.
> > >
> > > Since we directly use the results from Alemi et.al, the initial version did not have further justifications. However, in light of the reviewer’s concern, we have added text to the paper (Page. 4 Sec. 2.1) explaining further why one needs to set \beta < 1.
> > > Essentially, Alemi et.al. identify that the ELBO is comprised of the negative log-marginal likelihood term D, and the KL divergence term R, i.e. ELBO = -D -R. Further, they show the mutual information between the data “x” and the latent variable “z”, is bounded below by H-D and above by R (where H is a constant). Since the ELBO is equal to -D-R, the value of the ELBO on its own does not tell us about the mutual information between the observations and the latents, as different models (with say architectural differences) can achieve different D and R values for the same ELBO. Thus, Alemi et.al. prescribe setting $\beta<1$ for architectures with an auto-decoding behavior, so that we can get higher R values (by emphasizing on minimizing it less), pushing up the upper bound on mutual information I(x, z) achieved by the model
> > >
> > > ----------------------
> > > [R2] Could the authors provide technical details for this "argmax decoded programs"?
> > >
> > > We perform beam search to get an approximate solution for the argmax program given a question, as is standard practice [A]. This detail is added to Sec. 2.1. We did beam search instead of sampling since sequence models are known to suffer from a distributional mismatch between training and sampling [B],
> > > making beam search a conservative (and standard choice) in the literature for inference in sequence models [B]. For the module training stage, we find this leads to a good warm-start for the full objective (Eqn. 6) (which is optimized via. sampling).
> > >
> > > We also performed an experiment using sampling instead of beam search. In low-supervision settings, we find sampling leads to a drop in performance: with 10% supervision, accuracy on module training drops from 81.34 (+- 8.61) (with beam search) to 76.31 (+- 8.41) (with sampling) on validation. The performance with more supervision remains the same. We have added this finding to the paper (Page. 9, Results section).
> > >
> > > ----------------------
> > > [R2]  \gamma. The authors present the reason for that is because it is similar as Vedantam et al. (2018), which again totally points readers to check other references when the relaxation seems to have issue
> > >
> > > We use values of $\gamma > 1$ (see Appendix, and Page. 7) and thus for the case where $\beta > 1$ this still corresponds to a valid lower bound on the ELBO (for discrete-valued probability distributions, which is the case for answers in VQA). Thus, as such, we believe the relaxation does not have any issues (in addition to the justification already presented in the paper based on Vedantam et.al.).
> > >
> > > ----------------------
> > > [R2]  For objective (4), is \beta >=1 or < 1?
> > >
> > > \beta is set to 0.1 for all three stages, as clarified by the Algorithm box in the submitted version.
> > >
> > > References
> > > [A]: Vinyals, Oriol, Alexander Toshev, Samy Bengio, and Dumitru Erhan. 2014. “Show and Tell: A Neural Image Caption Generator.” arXiv [cs.CV]. arXiv. http://arxiv.org/abs/1411.4555.
> > > [B]: Bengio, Samy, Oriol Vinyals, Navdeep Jaitly, and Noam Shazeer. 2015. “Scheduled Sampling for Sequence Prediction with Recurrent Neural Networks.” arXiv [cs.LG]. arXiv. http://arxiv.org/abs/1506.03099.

---

> > > > ### Comment · AnonReviewer2 · 2018-11-28
> > > > **The presentation in the updated version is better**
> > > >
> > > > Thanks. I appreciate that the authors made improvement on the presentation based on reviews and provided detailed replies to my questions. The presentation in the updated version is better. As I understand, the paper used the existing score function estimator (Glynn, 1990) or REINFORCE (Williams, 1992) to solve the complex optimization part involving the KL divergence term. The optimization objective Eq. (6) is simplified to get the "warm starts" for relations between question x and program z, referred as question coding and for relation between program z, image i and answer a, referred as module training. For experiments, the authors compared V-NMN and NMN on the SHAPES dataset. The proposed method V-NMN is a probabilistic formulation. As I understand, the underlying graphical model provides interpretability for V-NMN. For the updated version, I think that the presentation is better, the probabilistic formulation V-NMN is interesting and the authors made lots of efforts to solve the optimization problem. On the other aspect, I am not very impressed by the technical part regarding creativity and insight. The experiment section is ordinary level. Overall, I think that the updated version is a good work and I updated my rating.
> > > >
> > > > I have a suggestion for the authors. It is nice to have a detailed introduction of the score function estimator (Glynn, 1990) or REINFORCE (Williams, 1992) and how they lead to the gradients updates, the step 3 and the step 10 in Algorithm 1 in Appendix. For the relation functions, such as those sequence to sequence models, most of them are discussed descriptively in the paper. It is nice to make the code open-source later or have mathematical discussion for them in the Appendix for reproducibility.

---

> > > > > ### Comment · Area_Chair1 · 2018-11-30
> > > > > **Please clarify your objection**
> > > > >
> > > > > I do not find the following statement particularly clear: "On the other aspect, I am not very impressed by the technical part regarding creativity and insight. The experiment section is ordinary level. Overall, I think that the updated version is a good work and I updated my rating."
> > > > >
> > > > > Please explain what you mean, as it is not obvious to me which part of your review underpins your assigning a score of 6. You are welcome to stand by your assessment, but it must be justified.

---

> > > > > > ### Comment · AnonReviewer2 · 2018-11-30
> > > > > > **Simple reply to AC**
> > > > > >
> > > > > > I am positive for this work after updated. I think that the probabilistic formulation of V-NMN is new. Technically, the authors need to solve a complex objective. As I mentioned in my above comments, I can see it is a hard question and the authors made lots of efforts to solve it. I think that the updated version is good now with more details where the original version is too descriptive. I do not think that the technical part is very impressive for me to give a higher score. It is common to simplify an complex objective to obtain "warm start" for some parameters. Another important part in the optimization is the KL divergence term. As I mentioned in my review, the authors used the existing score function estimator (Glynn, 1990) or REINFORCE (Williams, 1992). This part is not introduced enough in the paper, that is also the reason why I give a suggestion. If the authors can show me creativity, theoretical insight or illuminating explanation here, I may update my score. The reason why I point out "warm start" and KL divergence is because they are the key in the technical difficulty.
> > > > > >
> > > > > > For me, as I mention above, I think that the technical part is ordinary level since there is no enough creativity or theoretical insight. The experimental section is also an ordinary level, which just compares V-NMN and NMN on the SHAPES dataset. That is the reason why I give it a score 6. This paper also has a value that it provides a probabilistic formulation V-NMN to the VQA field. I am not experienced to judge the impact of a probabilistic model to this field. That is the other part which I may update my score if I see super-positive comments. I think that the technical part is relevant to my experience and I read it very carefully, and that is the reason why my score is more conservative.

---

> > > > > > > ### Comment · Area_Chair1 · 2018-11-30
> > > > > > > **Thanks**
> > > > > > >
> > > > > > > Thank you for clarifying, Reviewer 2. I will leave it to the authors to respond to this comment if they see fit.
> > > > > > >
> > > > > > > AC

---

> > > > > > > ### Author Response · Authors · 2018-12-01
> > > > > > > **Clarifying perspective on insight/novelty**
> > > > > > >
> > > > > > > We are glad R2 found the updated version of the paper more clear, and would like to thank the reviewer for prompt responses. As suggested by R2, we will release the code for our paper, along with all the settings to recreate the experiments on our github, and will add additional technical details of the REINFORCE estimator.
> > > > > > >
> > > > > > > Additionally, we would like to clarify for R2 what we think are the novel and original aspects of this work. While warm starting certain terms from previous stages is common practice, we still believe our key insight is not in terms of showing that warm-starts work, but in terms of realizing that one way to better capture the intent of a program specification is to model a stochastic latent space. This leads to better sharing of statistics across different questions and leads to a more meaningful latent program space. While this is well known for continuous valued latent variable (variational autoencoder style) models, this is relatively underexplored for the discrete, sequential program case.
> > > > > > >
> > > > > > > Overall, we believe our creativity and insight are not in terms of the mechanics or novelty of specific steps we undertook, but in terms of taking a concrete step towards probabilistic neural symbolic models which share statistics meaningfully in a latent space, and learn how to parse questions into programs as well as learn to execute programs using neural modules.
> > > > > > >
> > > > > > > A lot of important, open questions traditionally in AI have been in terms of representation learning, and modeling systematicity and compositional generalization [A], and we believe the line of work on neural-symbolic models (including ours as well as relevant works like [B, C]) are important steps towards solving these challenges.
> > > > > > >
> > > > > > > References
> > > > > > > [A]: Lake, Brenden M., and Marco Baroni. 2017. “Generalization without Systematicity: On the Compositional Skills of Sequence-to-Sequence Recurrent Networks.” arXiv [cs.CL]. arXiv. http://arxiv.org/abs/1711.00350.
> > > > > > > [B]: Yi, Kexin, Jiajun Wu, Chuang Gan, Antonio Torralba, Pushmeet Kohli, and Joshua B. Tenenbaum. 2018. “Neural-Symbolic VQA: Disentangling Reasoning from Vision and Language Understanding.” arXiv [cs.AI]. arXiv. http://arxiv.org/abs/1810.02338.
> > > > > > > [C]: Evans, Richard, David Saxton, David Amos, Pushmeet Kohli, and Edward Grefenstette. 2018. “Can Neural Networks Understand Logical Entailment?” arXiv [cs.NE]. arXiv. http://arxiv.org/abs/1802.08535.

---

> > > > > > > > ### Comment · AnonReviewer2 · 2018-12-01
> > > > > > > > **Thanks**
> > > > > > > >
> > > > > > > > I'd like to thank the authors for the comments. I agree with your points. I think that the updated version is a good work and I give it a positive score. As I pointed out, I can see that this work proposes a probabilistic neural symbolic models to the field of VQA. The probabilistic formulation is popular in many fields in machine learning, which I do not see it as special compared to non-probabilistic models. I can not strongly support a work only because it is formulated as a probabilistic model. As I mentioned, I do not think that I am experienced enough to determine the impact of this probabilistic model to the field of VQA, so I may underestimate the importance.
> > > > > > > >
> > > > > > > > I also think that a work should be evaluated from different aspects. As I mentioned, I focus more on technical section compared to other reviewers. I'd like to clarify again that my creativity and insight comments are about technical section.
> > > > > > > >
> > > > > > > > Thank the authors for detailed replies to my concerns. I think that the updated version is good and I give this work a positive rating. We both make some clarification. I do not see a contradiction of points between the authors and me which we need to clarify further. I think that our discussion is clear for the AC. As the authors pointed out several times, how we view the importance of a probabilistic model to the field of VQA is crucial to determine the impact of this work. For this type of question, I can not change my support point to a strongly support point from a discussion with authors. As I mentioned, I do not think that I am experienced enough to determine the impact of this probabilistic model to the field of VQA. For technical section and experiment section, I still think that they are ordinary level to me.
> > > > > > > >
> > > > > > > > Thank the authors. AC asked me to clarify my review. If we do not see a need to clarify further, we can leave it to AC and see if the AC still have questions. I think that our discussion is clear and we've discussed a lot of details about this work. I appreciate that.

---

> > > ### Author Response · Authors · 2018-11-27
> > > **Addressing R2's concerns [Part 2/2]**
> > >
> > > ----------------------
> > > [R2] Why the objective (1) is hard to train but objective (4) is possible to train?
> > >
> > > Objective (4) is Objective (6) in the updated version, and we refer to that in the discussion below.
> > >
> > > As mentioned in the submission (Page. 4, Sec. 2.1), objective (6) is possible to train because it uses ‘warm-starts’ from the other two stages of training (question coding, Eqn. 1, and module training, Eqn. 5 respectively). We provide further intuition for why objective (6) is inherently difficult to train: essentially the parameterization of p(a|i,z) is done by assembling neural module networks on the fly based on the predicted program (z), which is then trained using SGD. Thus, the optimization landscape is in some sense discontinuous in the parameters of the modules (since a different set of modules, with different parameters, could be chosen based on the program). Hence, optimizing Eqn. 6 from scratch is hard (without a good inference network q(z| x) and a good parameterization/ initialization of p(a|i,z)).
> > >
> > > ----------------------
> > > [R2] Other related work:
> > > 1] Interpretable Visual Question Answering by Visual Grounding from Attention Supervision Mining (Zhang et al. 2018)
> > > -- Our work focuses on a different notion of interpretability for VQA compared to Zhang et.al. While we are interested in a notion of interpretability that preserves a notion/ syntactic specification of `how' to answer a question, this paper is interested in grounding the answer into appropriate regions in the image. This is an orthogonal notion of interpretability compared to what we are mainly interested in; in the sense that we are explicitly interested in question answering via human-interpretable programs, while Zhang et.al. is interested in grounding answers into relevant regions in an image.
> > >
> > > 2] Neural-Symbolic VQA: Disentangling Reasoning from Vision and Language Understanding (Yi et al. 2018).
> > > -- This is certainly very relevant work, thank you for the pointer. Conceptually, at a high level, the goals of this work and ours are similar: both the approaches want to do question answering with limited question program supervision. While Yi et al. seem to approach this problem by simplifying the p(a|i,z) mapping (in context of our model) by first converting the image into a symbolic table, we approach this goal by modeling a stochastic latent space. In some sense Yi et al., and our approaches are orthogonal, as one can use our probabilistic latent space in conjunction with this work., and thus we believe our approach is of independent interest.
> > >
> > > Further, our approach is addressing the full complexity of learning such neural-symbolic models in an end to end manner, where program execution is being learned in conjunction with parsing questions into programs. This is arguably more general, as parsing images can into tables need not be a sufficient representation for visual recognition across different settings. We have added a discussion on the differences to Yi et.al in related work.

---

### Official Review · AnonReviewer1 · 2018-11-03
**Nice paper, well written and through evaluation**

**Rating:** 8
**Confidence:** 5

**Review:**

This paper proposes a discrete, structured latent variable model for visual question answering that involves compositional generalization and reasoning. In comparison to the existing approach, this paper well addressed the challenge of learning discrete latent variables in the presence of uncertainty. The results show a significant gain in performance as well as the capability of the model to generalize composition program to unseen data effectively. The qualitative analysis shows that the proposed model not only get the correct answer but also the correct behavior that leads to the answer.

---

> ### Comment · Area_Chair1 · 2018-11-07
> **A bit more detail**
>
> Thank you for your review. It is quite short, so it would be good to expand upon what you think makes this paper, perhaps in the form of a discussion with Reviewer 2, whose score is significantly different from yours.

---

> ### Comment · Area_Chair1 · 2018-11-15
> **Urgent need for detail**
>
> Can I please ask Reviewer 1 to, with due urgency, expand upon their review and/or comment upon those of the other reviewers, so as to proffer an appropriate defence of their recommendation in favour of the paper DURING the discussion period.

---

### Public Comment · (anonymous) · 2018-11-19
**Prior work on VAEs with discrete latent program space**

This is a nice submission of approaching visual question answering using a probabilistic neural-symbolic model with discrete latent program space. However, there has been prior work on probabilistic neural-symbolic models with a latent program space (Yin et al., 2018), which seems to be one of the major contributions claimed in this submission (at least from the TL;DR line :)) I was wondering if the authors could explain the difference between V-NMN and Yin et al., which would better substantiate the novelty of this work compared with Yin et al.

Reference:
[1] Pengcheng Yin, Chunting Zhou, Junxian He, Graham Neubig. StructVAE: Tree-structured Latent Variable Models for Semi-supervised Semantic Parsing. ACL 2018.

---

> ### Author Response · Authors · 2018-11-27
> **Relevant Work, Updates to paper to discuss differences.**
>
> Thank you for pointing us to this highly relevant work! The approach is indeed quite relevant but has some key differences which we highlight now in the updated draft (in related work).
>
> In Yin et.al., while the programs are modeled as a latent variable, the model does not capture ``how'' to execute the programs that it generates, and indeed the tasks considered only have a notion of ``parsing’’ into programs but not program execution. More specifically, the model presented in Yin et.al. is a unimodal model with a structured latent space, where the observed modality is the raw text/ question. However, in our model, we have a second modality which is the output of what gets executed when the program runs, and we capture both jointly in our model. Thus we argue that we address probabilistic neural-symbolic learning in a more general setting: where one has to parse a question into programs *as well as* learn to execute them by training neural modules.
> However, the key idea of a tree-structured syntactic latent space to represent programs from the work is very interesting and would be relevant to use in our model as well.

---

### Author Response · Authors · 2018-11-27
**Updates to paper addressing R2's concerns, specific replies to R2 and Anon.**

We thank the reviewers for the detailed comments and questions, and are encouraged reviewers found the paper well written [R3], our approach worth studying [R3] and that we address the problem well [R1]. In the revised version, we have highlighted the sections in magenta which address the concerns of R2 but have not changed since the initial submission, for ease of reference. Changes we made since the initial submission are marked in blue.

Please note that whenever we refer to text in the paper in this discussion, we are referring to the updated version of the paper (and not the submitted version, although we might point to text explicitly present in the submitted version based on the above color scheme).

To reiterate, the goal of this work is not just to get higher empirical performance on VQA. Instead, our aim is to augment an existing class of techniques -- that has been shown to have desirable properties like interpretability and compositionality (Johnson et.al., Hu et.al.) -- with a probabilistic treatment. Concretely and in the short term, this results in higher performance for capturing the intent of human program specifications better (via. better semi-supervised learning), but arguably equally importantly, in the longer term, this is a framework for building probabilistic, neural-symbolic models. Note that neural-symbolic models bring together the power of deep representation learning with the systematic generalization capabilities of symbolic reasoning, promising the best of both worlds. Moreover, probabilistic tools provide a systematic and flexible framework for modeling and inference in general. With these high-level goals, we generally restrict the comparison to previous VQA approaches which use explicit program representations and learn to execute them.

Comments addressing specific issues with Sec. 2.1 are in the replies to R2 and more specific comparisons to closely related work follow in replies to R2 and the Anonymous comment.
We thank R3 for pointing out typos other writing-related issues, which we address in the updated version.

---

### Public Comment · (anonymous) · 2019-02-03
**Results on other datasets**

The authors report their results on SHAPES dataset. In visual reasoning, however, the CLEVR dataset is a much more acceptable benchmark. Is there any specific reason that the authors don't use CLEVR dataset in spite of referring to all the papers in NMN series (Johnson et al., 2017; Hu et al., 2017; Andreas et al., 2016a) ?

---

### Meta-Review · Area_Chair1 · 2018-12-14
**Very borderline**

**Confidence:** 3
**Recommendation:** Reject

**Metareview:**

This paper proposes a latent variable approach to the neural module networks of Andreas et al, whereby the program determining the structure of a module network is a structured discrete latent variable. The authors explore inference mechanisms over such programs and evaluate them on SHAPES.

This paper may seem acceptable on the basis of its scores, but R1 (in particular) and R3 did a shambolic job of reviewing: their reviews are extremely short, and offer no substance to justify their scores. R2 has admirably engaged in discussion and upped their score to 6, but continue to find the paper fairly borderline, as do I. Weighing the reviews by the confidence I have in the reviewers based on their engagement, I would have to concur with R2 that this paper is very borderline. I like the core idea, but agree that the presentation of the inference techniques for V-NMN is complex and its presentation could stand to be significantly improved. I appreciate that the authors have made some updates on the basis of R2's feedback, but unfortunately due to the competitive nature of this year's ICLR and the number of acceptable paper, I cannot fully recommend acceptance at this time.

As a complete side note, it is surprising not to see the Kingma & Welling (2013) VAE paper cited here, given the topic.